# Evaluation of *Ascophyllum nodosum* extract supplementation on feed degradability, ruminal fermentation and methane production using the rumen simulation technique (RUSITEC)

Aaron Casey[ID][1], Tommy M. Boland[1], Alan K. Kelly[1], Zoe C. McKay[1], Maria Markiewicz-Keszycka[1], Armin Mirzapour-Koudash[1], Marco Garcia-Vaquero[1], Sivaprakash Balakrishnan[2], Kieran J. Guinan[ID][2], John T. O'Sullivan[2], Stafford Vigors[ID][1]*

1 School of Agriculture and Food Science, University College Dublin, Belfield, Dublin 4, Ireland,
2 BioAtlantis Ltd, Clash Industrial Estate, Tralee, Co. Kerry, Ireland

* staffordvigors1@ucd.ie

## Abstract

This study investigated the effects of liquid *Ascophyllum nodosum* (ASC) extracts on feed degradability, ruminal fermentation, total gas and methane ($CH_4$) production using the rumen simulation technique (RUSITEC). Two experiments were conducted: Experiment 1 assessed a grass-based diet with an ASC extract included at 0.1%, 0.2% and 0.3% of dry matter (DM), while Experiment 2 evaluated a grass silage-based diet with an ASC extract included at 0.3%, 0.4% and 0.5% of DM. Both diets included a control treatment (CTR; no seaweed extract). In Experiment 1, the 0.3% treatment reduced $CH_4$ production relative to CTR, primarily by decreasing nutrient degradation and total volatile fatty acid production, resulting in reductions of 20% for total gas, 34% for $CH_4$ (mmol/d) and 30% for $CH_4$ (mmol/g of digestible organic matter; DOM). Metabolic hydrogen produced and incorporated were also reduced. In Experiment 2, the 0.5% treatment decreased DM and organic matter degradation compared to CTR, with no effects on crude protein or neutral detergent fibre degradation, or rumen fermentation parameters. Despite this, all ASC extract treatments reduced total gas by 20–24%, $CH_4$ (mmol/d) by 23–26% and $CH_4$ (mmol/g DOM) by 21–29% relative to CTR. Metabolic hydrogen recovery decreased by 22% across all ASC extract treatments, suggesting other mechanisms were at play redirecting hydrogen away from methanogenesis. These findings highlight the potential of ASC extract as a natural $CH_4$-mitigating feed additive under controlled fermentation conditions, supporting the development of more sustainable ruminant production systems.

## Introduction

Agriculture is responsible for the production of 37.8% of Irish greenhouse gas (GHG) emissions, with methane ($CH_4$) from ruminal fermentation and manure management

**Data availability statement:** All relevant data are within the manuscript and its Supporting information files.

**Funding:** The project was supported by Enterprise Ireland in the form of a grant awarded to TB (IP/2021/0973).

**Competing interests:** The authors have declared that no competing interests exist.

contributing up to 72.1% of agricultural emissions [1]. Through ruminal fermentation, feed is degraded by microorganisms to produce volatile fatty acids (VFA), hydrogen ($H_2$) and carbon dioxide ($CO_2$). The VFA's are absorbed mainly through the rumen wall and provide energy to the animal, however, $H_2$ is toxic to the rumen environment, and therefore must be removed to ensure the fermentation process continues [2]. Methanogenesis is considered the main metabolic hydrogen ([H]) sink in the rumen whereby methanogenic archaea reduce $CO_2$ to $CH_4$ by utilizing $H_2$ [3,4], causing an energy loss of 2–12% of gross energy intake [5]. The European Green Deal aims to reduce GHG emissions by 55% by 2030 compared to 1990 levels [6]. Strategies to achieve these targets include exploring anti-methanogenic feed additives that specifically target methanogenesis and reduce $CH_4$ emissions. Feed additives can reduce $CH_4$ by inhibiting $H_2$ production, providing an alternative H sink, or suppressing methanogenic archaea, with diet and microbiome modifications serving as potential $CH_4$ mitigation tools [7–9].

Previous research has explored dietary feed additives, including ionophores, chemical compounds, essential oils, and plant secondary compounds, to mitigate $CH_4$ emissions [7,10]. Seaweeds have promise for $CH_4$ mitigation [11,12], particularly red seaweed species like *Asparagopsis spp.*[13,14]. However, the non-native status of this seaweed in Ireland, along with the environmental and safety concerns related to its halogenated compound bromoform, require the exploration of alternative seaweed species. *Ascophyllum nodosum* (ASC), a brown seaweed native to the North Atlantic and abundantly available along the Irish coastline, offers a sustainable alternative. Unlike *Asparagopsis*, ASC does not contain bromoform but still possesses anti-methanogenic potential due to its high phlorotannin (PT) content. Phlorotannins are similar to terrestrial tannins and have the ability to bind to proteins and carbohydrates [15], thereby lowering nutrient degradation. They also exert strong anti-microbial activity, particularly against fibre-degrading bacteria, which can subsequently decrease methanogenesis [16]. Although reduced nutrient degradation may limit feed utilization, a recent meta-analysis by Sofyan [12] reported that PT-containing brown seaweeds can modulate rumen fermentation by altering microbial populations and VFA profiles, ultimately lowering $CH_4$ production without detrimental effects on animal performance. These mechanisms are consistent with *in vitro* findings, where ASC supplementation reduced total gas production (TGP), $CH_4$ production [17,18], nitrogen (N) degradation [19], and ammonia-N concentration in rumen fluid [20].

While many seaweed species are often used in an unprocessed form in animal feeds, the effects of a targeted extraction of potential bioactive components on nutrient utilization and methanogenesis remain relatively unknown. To address this, Wang [21] tested different PT concentrations from ASC in an *in vitro* batch culture system, revealing that PT readily formed complexes with proteins, leading to a linear reduction in fermentation and $CH_4$ production. However, batch cultures face challenges in maintaining a stable fermentation over multiple days due to the build-up of fermentation products [19]. The rumen simulation technique (RUSITEC), designed by Czerkawski and Breckenridge [22], offers a more effective alternative by allowing for prolonged and consistent *in vitro* fermentation, making it well-suited for screening anti-methanogenic feed additives before conducting costly, time-consuming, and labour-intensive *in vivo* studies [23].

The novelty of this study was to test liquid ASC extracts *in vitro* that could potentially pave the way for future inclusion *in vivo* via water-based delivery. The hypothesis of this study was that supplementing PT-rich liquid ASC extracts would mitigate $CH_4$ production *in vitro* by decreasing substrate degradability and TGP, thereby lowering $H_2$ availability for methanogenesis and demonstrating their potential for effective *in vivo* $CH_4$ mitigation. The objective of this study was to evaluate the effects of an ASC extract on *in vitro* nutrient degradation, ruminal fermentation and $CH_4$ production to determine the inclusion level with the strongest anti-methanogenic effect for future application in animal studies.

## Materials and methods

### Experimental licensing

This experiment was conducted at UCD Lyons Research Farm, Celbridge, Co. Kildare, Ireland, W23 ENY2 ($53^0$ 17' 56" N, $6^0$ 32' 18" W). Rumen inoculum during Experiment 2 was harvested from four rumen cannulated Aberdeen Angus x Friesian heifers at Teagasc Grange Animal and Bioscience Research Department, Co. Meath. Animals utilised for the collection of rumen inoculum were cared for in accordance with the Teagasc Animal Ethics Committee. All procedures were approved by the Health Product Regulatory Authority (AE19132/P113), conducted under the European Directive 2010/63/EU and S.I. No. 543 of 2012.

### Experimental design and apparatus

The study consisted of two separate complete randomized design 17-d experiments, the initial 10 days dedicated to microbial adaptation and fermentation stabilisation, as outlined by Jaurena [24], and the remaining 7 days for data collection and sampling. Each experiment contained two RUSITEC systems with six vessels per system (Sanshin Industrial Co. Ltd, Yokohama, Japan), used to simulate the rumen environment *in vitro*. Each vessel had a nominal volume of 800 ml and the general incubation procedure was carried out as described by Czerkawski and Breckenridge [22].

### Diets and treatments

Different basal diets were employed for each experiment, with all vessels receiving 20g dry matter (DM) of feed components—comprising of two separate nylon feed bags (100 – μm pore size; 5 x 10 cm – concentrate; 10 x 20 cm – forage; ANKOM ™ Technology, Macedon, NY, USA); one forage bag containing 16 g of forage DM and one concentrate bag containing 4 g of concentrate DM. The forage component of the basal diet offered to the donor animals at the time of rumen inoculum collection was subsequently then utilized as the forage component for each respective *in vitro* experiment. This approach aimed to minimise time required for microbial adaptation to the experimental diets while also allowing assessment of the $CH_4$-mitigating effects of ASC supplementation across forages differing in fermentability. For Experiment 1, a perennial ryegrass (*Lolium perenne*) based pasture was utilised, for Experiment 2, a perennial ryegrass-based grass silage was utilized. The feed bags were provided on an 80:20 F:C DM basis. Prior to inclusion in the vessels, both concentrate and forage components underwent a drying process in a forced air oven at 55°C for 48 h. The concentrate feed component was ground through a 1 mm sieve, and the forage component was chopped to 1–2 cm in length using a bowl chopper. The chemical composition of the forages and concentrate is presented in Table 1.

In Experiment 1, rumen inoculum was obtained post slaughter from three Aberdeen Angus x Holstein Friesian steers (320 kg body weight; 15 months of age) in an abattoir, where all animals had been offered fresh grass and concentrate prior to slaughter. In Experiment 2, rumen inoculum was obtained from four ruminally cannulated Aberdeen Angus x Holstein Friesian steers (270 kg bodyweight; 12 months of age) before the morning feeding at 0800 h, where all animals had been offered grass silage *ad libitum* plus 3 kg concentrate on a fresh weight basis three weeks prior to the experiment.

The seaweed extracts were manufactured by BioAtlantis Ltd. (Clash Industrial Estate, Tralee, Co. Kerry, Ireland) using the brown seaweed ASC, which was hand harvested in Ireland during the winter season. The process involved aqueous extraction under conditions of high temperature and pH, followed by pH adjustments, filtration and evaporation to reach

**Table 1. Chemical composition (g/kg DM unless otherwise stated) of forage and concentrate for Experiment 1 and 2.**

| Parameter | Experiment 1 | | Experiment 2 | |
|---|---|---|---|---|
| | Perennial ryegrass pasture | Concentrate | Grass Silage | Concentrate |
| DM (g/kg) | 216 | 894 | 532 | 892 |
| Ash | 78.7 | 53.7 | 86.0 | 46.9 |
| CP | 129 | 159 | 82.6 | 115 |
| NDF | 379 | 199 | 509 | 186 |
| ADF | 196 | 75.1 | 303 | 93.8 |
| OM | 921 | 946 | 914 | 951 |
| Starch | . | 498 | . | 448 |
| GE (MJ/ kg DM) | 17.5 | 17.5 | 17.1 | 17.0 |
| EE | 30.5 | 26.3 | 11.7 | 19.3 |

DM, dry matter; CP, crude protein; NDF, neutral detergent fibre; ADF, acid detergent fibre; OM, organic matter; GE, gross energy; EE, ether extract.

approximate 50% w/v levels. The extract typically contains 12–16% inorganic matter and 25–30%, organic matter (OM), the latter including approximately 1.5–2.5% mannitol and 7–10% PT on a DM basis, in addition to alginic acid polymers, fucose and glucan polymers. Extracts differed slightly in chemical composition between experiments, with PT concentrations of 163.2 mg PGE/g in Experiment 1 and 142.5 mg PGE/g in Experiment 2.

In Experiment 1, treatments consisted of three seaweed extract inclusion rates of 20 mg/ 20g DM (**0.1%**), 40 mg/ 20g DM (**0.2%**) and 60 mg/ 20g DM (**0.3%**). In Experiment 2, the extract was supplied at 60 mg/ 20g DM (**0.3%**), 80 mg/ 20g DM (**0.4%**) and 100 mg/ 20g DM (**0.5%**). Both experiments included a control diet which received no seaweed extract (**CTR**). Treatments were replicated three times and randomly allocated to each vessel. The chemical composition of the seaweed extract is included in Table 2.

## Experimental procedure

On d-0, rumen inoculum (fluid and digesta) was collected, strained through four layers of cheesecloth, pooled, flushed with $CO_2$, and promptly incubated at 39°C. Within 60 min of collection, the inoculated mixture was transferred to the RUSITEC vessels. Each vessel received 450 ml of rumen inoculum and 350 ml of artificial saliva [25]. Dietary treatments were introduced in nylon bags (100-µm pore size), including 70 g of rumen solid digesta, a second bag containing the 16 g forage DM, and a third bag containing 4 g of concentrate DM.

Artificial saliva, prepared daily, was continuously infused at a rate of 640 ml/d (dilution rate of 3.33%/h) using a peristaltic pump to prevent the washout of rumen microbes. Effluent and fermentation gasses from each vessel were collected into effluent bottles and gas collection bags, respectively. After 24 hours at 0900 h, each vessel was opened, and two

**Table 2. Chemical composition of the *Ascophyllum nodosum* extract for Experiment 1 and 2. Figures presented are mean values±standard deviation.**

| Parameter | Experiment 1 | Experiment 2 |
|---|---|---|
| DM (g/kg) | 399±2.7 | 415±2.2 |
| Ash (g/kg) | 163±1.1 | 157±0.9 |
| Total tannin content (mg ChE/g) | 6.6±0.03 | 6.5±0.05 |
| Total soluble carbohydrates (mg GlcE/g) | 10.8±0.41 | 20.3±0.50 |
| Total phenolic content (mg GAE/g) | 272±0.0 | 202±0.9 |
| Total PT content (mg PGE/g) | 163±0.2 | 142±0.7 |

of the initial three bags containing rumen digesta solids, along with the bag containing the concentrate, were removed, squeezed, and washed in 50 ml of artificial saliva. Liquid ASC extracts were pipetted into the vessel daily at the designated dosage rate on a single occasion. The liquid fractions of the washings in artificial saliva were returned to the vessels, and two new nylon bags, containing 16 g of forage DM and 4 g of concentrate DM, were inserted. On subsequent days, the nylon bag, containing the forage that had been in the vessel for 48 h was replaced with a new forage bag, and the nylon bag containing the concentrate, which had been in the vessel for 24 h, was similarly replaced.

## Data collection

Dry matter degradation, vessel pH, overflow container pH, TGP and $CH_4$ percentage of TGP were continuously measured throughout the experiment. On d 11–17, a 4 ml sample was collected from the overflow containers with the pH immediately measured using a digital pH meter (Phoenix Instrument EC-25 Ph/Conductivity Portable Meter). This sample was then acidified using 1 ml of 50% trichloroacetic acid and stored at −20°C for VFA analysis.

Nylon bags, after collection, were rinsed with cold water, and feed residues were washed in a domestic washing machine using the cold rinse cycle without detergent (30 min) to remove loosely attached bacteria. The feed residues were then dried in a 55°C forced air oven for 48 h and weighed. Feed dry matter degradation was calculated based on the material disappearance from the nylon bags after 24 h and 48 h of incubation for concentrates and forage, respectively. Chemical composition of the dried incubation residues was determined to calculate the degradation of feed components.

Total gas production was measured using reusable polyethylene gas bags fitted with one-way valves. Bags were attached to overflow containers where the gas was collected and measured using a DC dry gas test meter (Shinagwa Corp.; Tokyo, Japan). From here, the $CH_4$ percentage was analysed using an infra-red ADC SB2000 wall mounted analyser (ADC Gas Analysis; Hoddeston, UK). Daily calibrations of the ADC SB2000 were conducted with a 10% $CH_4$ span gas. The $CH_4$% and TGP were then used to calculate $CH_4$ (mmol/d) using the following formula:

$$CH_4 \left( mmol\ d^{-1} \right) = \frac{\left[ CH_4(\%) \times TGP\ \left( L\ d^{-1} \right) \right] \times 0.656}{16.04} \times 1000$$

Where $CH_4\% \times TGP$ (total gas production) gives the volume of $CH_4$ (L/d), 0.656 g/L is the density of $CH_4$ and 16.04 g/mol is the molecular weight of $CH_4$, and the factor 1000 converts mol to mmol.

## Chemical analysis

**Feed analysis.** The DM content of the feed samples was determined by drying the samples at 105 °C for 16 h [26a; method 960.15]. Ash concentrations were determined through complete combustion in a muffle furnace (Nabertherm, GmbH, Lilienthal, Germany) at 550 °C for 4 h [26b; method 942.05]. Nitrogen concentrations of the feed were assessed using a LECO FP 528 instrument (Leco Instruments, UK, Ltd, Stockport, UK), and the resulting values were multiplied by 6.25 to determine crude protein (CP) concentrations [26c; method 990.03]. Neutral detergent fibre (NDF) and acid detergent fibre (ADF) concentrations were determined following the method of Van Soest [26] using the ANKOM220 Fibre Analyzer (ANKOM Technology, Macedon, NY). Forage and concentrate samples were analysed for NDF with sodium sulphite, with heat stable amylase included for concentrate samples only. The starch content of feed was determined using the Megazyme Total Starch Assay Procedure (product no. K-TSTA, Megazyme International Ltd., Wicklow, Ireland). Gross energy (GE) content of the feed was determined by bomb calorimeter (Parr 1281 Bomb Calorimeter, Parr Instrument Company, Moline, IL). Ether extract (EE) of feed samples was determined using a Soxtec instrument (tecator) according to the method and light petroleum ether.

**Seaweed analysis.** Total soluble carbohydrates were determined by the phenol-sulphuric acid method following the protocol as described by [27]. Briefly, 100 µL of test samples or standard (D-glucose, 50–250 mg/L) were mixed

thoroughly with 100 μL of 0.8% (w/v) phenol solution and 2 ml of sulphuric acid (95–98%). The mixtures were allowed to stand for 10 min and then incubated in a water bath at 30 °C for 20 min. Absorbance was measured at 490 nm using a spectrophotometer (VICTOR® Nivo™, PerkinElmer, Pontyclun, United Kingdom), and results were expressed as mg glucose equivalents (GlcE) per g of fresh sample.

Total phenolic content and total PT content were determined as described by Garcia-Vaquero [28]. Briefly, 100 μL of sample or standard (gallic acid for total phenolic content or phloroglucinol for total PT content, 25–300 mg/L) was mixed with 2 mL of sodium carbonate solution (2% w/v). After 2 min, 100 μL of Folin–Ciocalteu's solution (1 M) was added, and the mixtures were incubated in dark conditions (30 min, room temperature). Absorbance was measured at 720 nm in a spectrophotometer (VICTOR® Nivo™, PerkinElmer, Pontyclun, United Kingdom). Results of total phenolic content and total PT content were expressed as mg gallic acid equivalents (GAE) and mg phloroglucinol equivalents (PGE) per g of fresh sample, respectively.

Total tannin contents were determined as described by Liu [29]. A volume of 50 μL of sample or standard (catechin, 15–150 mg/L) were mixed thoroughly with 1.5 ml of methanolic vanillin solution (4% w/v) and 750 μL of 37% hydrochloric acid. The mixtures were incubated in dark conditions (20 min, room temperature) and the absorbance was read at 500 nm using in a spectrophotometer (VICTOR® Nivo™, PerkinElmer, Pontyclun, United Kingdom). Results of total tannin content were expressed as mg catechin equivalents (ChE) per g of fresh sample.

**VFA analysis.** Liquid samples, previously collected from the overflow containers, acidified, and frozen pending analysis, were thawed overnight in a fridge at 4 °C and then centrifuged at $1{,}800 \times g$ for 10 min at 4 °C. Following this, 250 μl of supernatant was diluted with 3.75 ml of distilled water and 1 ml of an internal standard solution (0.5808 g 3-methylvaleric acid in 1,000 mL of HPLC grade water). The resulting solution underwent centrifugation at $260 \times g$ for 5 min at room temperature and was then filtered through a syringe tip filter (polytetrafluoroethylene, 25 mm diameter, 0.45 μm) into 2 ml gas chromatography vials, with each sample analysed in duplicate. The concentration of VFA's was determined by gas chromatography (Scion 456-GC, Scion Instrument, Scotland, UK) fitted with a DB-FFAP capillary column (15 m × 0.53 mm: 1.00 μm, Agilent Technologies, USA). Metabolic hydrogen produced, incorporated and recovered was accounted from the VFA production based off the calculation by Ungerfeld [30] without the inclusion of $H_2$.

## Statistical analysis

Data was checked for normality and homogeneity of variance by histograms, qq plots, and formal statistical tests as part of the UNIVARIATE procedure of SAS® studio (edition 3.81). Data was analysed using the PROC MIXED procedure, with fermentation vessel included as a random effect. The model included the fixed effects of treatment and day. Differences between treatments were determined by F-tests using Type III sums of squares. Pairwise comparisons between treatment means were evaluated using the PDIFF option of the LSMEANS statement with Tukey adjustment in PROC MIXED. Statistically significant difference was assumed at $P < 0.05$ and a tendency toward significance was assumed at $P \geq 0.05$ but $P < 0.10$.

## Results

### Diet degradation

The degradation of DM, OM, CP and NDF in Experiment 1 and 2 are presented in Tables 3 and 4, respectively. In Experiment 1, DM, OM and CP degradation were reduced for treatment 0.3% relative to CTR and 0.1% treatments, while NDF degradation was reduced for treatment 0.3% versus CTR only ($P < 0.05$). In Experiment 2, the 0.5% treatment reduced DM and OM degradation by 5% compared to CTR ($P < 0.05$). There was no effect of treatments on CP or NDF degradation ($P > 0.05$).

Table 3. Mean effect of treatments on nutrient degradation (g/kg DM unless stated) in Experiment 1.

| Parameter | Treatments | | | | SEM | P-value |
|---|---|---|---|---|---|---|
| | CTR | 0.1% | 0.2% | 0.3% | | |
| DM (g/kg) | 866[a] | 865[a] | 858[ab] | 818[b] | 13.2 | 0.010 |
| OM | 864[a] | 863[a] | 856[ab] | 814[b] | 11.8 | 0.011 |
| CP | 884[a] | 889[a] | 878[ab] | 853[b] | 8.02 | 0.011 |
| NDF | 754[a] | 742[ab] | 720[ab] | 642[b] | 29.1 | 0.040 |

[a-b]Within each column, treatment means not sharing the same superscript differ (P<0.05).

CTR=0 mg/ 20g DM of *Ascophyllum nodosum* (ASC) extract; 0.1%=20 mg/ 20g DM of ASC extract; 0.2%=40 mg/ 20g DM of ASC extract; 0.3%=60 mg/ 20g DM of ASC extract.

DM, Dry matter; OM, Organic matter; CP, Crude protein; NDF, Neutral detergent fibre.

Table 4. Mean effect of treatments on nutrient degradation (g/kg DM unless stated) in Experiment 2.

| Parameter | Treatments | | | | SEM | P-value |
|---|---|---|---|---|---|---|
| | CTR | 0.3% | 0.4% | 0.5% | | |
| DM (g/kg) | 673[a] | 656[ab] | 657[ab] | 638[b] | 7.90 | 0.035 |
| OM | 658[a] | 639[a] | 641[ab] | 621[b] | 8.13 | 0.032 |
| CP | 933 | 923 | 916 | 920 | 9.65 | 0.624 |
| NDF | 319 | 315 | 319 | 291 | 13.8 | 0.434 |

[a-b]Within each column, treatment means not sharing the same superscript differ (P<0.05).

CTR=0 mg/ 20g DM of *Ascophyllum nodosum* (ASC) extract; 0.1%=20 mg/ 20g DM of ASC extract; 0.2%=40 mg/ 20g DM of ASC extract; 0.3%=60 mg/ 20g DM of ASC extract.

DM, Dry matter; OM, Organic matter; CP, Crude protein; NDF, Neutral detergent fibre.

## Fermentation variables

The effect of ASC extract on rumen fermentation and VFA production for Experiment 1 and 2 are presented in Tables 5 and 6, respectively. In Experiment 1, total VFA and butyrate production were reduced for treatment 0.3% compared to all other treatments (P<0.05), while acetate production was reduced relative to treatment 0.1% only (P<0.05). There was no effect of treatments on the production of propionate, iso-butyrate, valerate and iso-valerate along with acetate to propionate ratio (P>0.05). Metabolic hydrogen produced was reduced for treatment 0.3% relative to both CTR and 0.1%, while [H] incorporated was reduced for treatment 0.3% versus CTR only (P<0.05). There was no effect of treatment on [H] recovered or vessel pH (P>0.05).

In Experiment 2, there was no difference between treatments for any VFA production parameters, [H] produced, incorporated or vessel pH (P>0.05). Metabolic hydrogen recovered was reduced for all three inclusion rates by 22% relative to CTR (P<0.05).

## Total gas and methane parameters

Tables 5 and 6 report the effect of ASC extract on TGP (L/d) and $CH_4$ parameters (mmol/d, and $CH_4$ mmol/g digestible organic matter; DOM) for Experiment 1 and 2, respectively. In Experiment 1, the 0.3% treatment reduced TGP by 20%, $CH_4$ (mmol/d) by 34% and $CH_4$ (mmol/g DOM) by 30% compared to CTR (P<0.05). In Experiment 2, all three inclusion rates of ASC extract (0.3%, 0.4% and 0.5%) reduced TGP by 20–24%, $CH_4$ (mmol/d) by 23–27% and $CH_4$ (mmol/g DOM) by 21–29% compared CTR (P<0.05).

**Table 5. Mean effect of treatments on gas production and rumen fermentation parameters in Experiment 1.**

| Parameter | Treatments | | | | SEM | *P*-value |
|---|---|---|---|---|---|---|
| | CTR | 0.1% | 0.2% | 0.3% | | |
| TGP (L/d) | 3.12[a] | 3.00[ab] | 3.05[ab] | 2.50[b] | 0.158 | 0.035 |
| $CH_4$ (mmol/d) | 14.2[a] | 12.3[ab] | 12.9[ab] | 9.3[b] | 1.13 | 0.029 |
| $CH_4$ (mmol/g DOM) | 0.84[a] | 0.73[ab] | 0.76[ab] | 0.58[b] | 0.062 | 0.037 |
| Vessel pH | 6.32 | 6.37 | 6.32 | 6.47 | 0.063 | 0.284 |
| **VFA production (mmol/d)** | | | | | | |
| TVFA | 70.1[a] | 70.3[a] | 70.7[a] | 58.1[b] | 2.75 | 0.022 |
| Acetate | 33.6[ab] | 34.9[a] | 34.3[ab] | 27.2[b] | 1.63 | 0.026 |
| Propionate | 16.3 | 16.4 | 17.1 | 14.5 | 0.72 | 0.161 |
| Butyrate | 14.5[a] | 13.7[a] | 14.0[a] | 11.3[b] | 0.57 | 0.007 |
| Iso-butyrate | 0.55 | 0.54 | 0.55 | 0.46 | 0.032 | 0.231 |
| Valerate | 3.07 | 2.58 | 2.84 | 2.69 | 0.170 | 0.210 |
| Iso-valerate | 2.04 | 2.01 | 2.01 | 1.92 | 0.109 | 0.872 |
| Acetate: propionate ratio | 2.07 | 2.12 | 2.02 | 1.87 | 0.078 | 0.192 |
| **Estimated H parameters (mmol/d)** | | | | | | |
| [H] production | 141.7[a] | 141.4[a] | 141.7[a] | 114.7[b] | 5.66 | 0.012 |
| [H] incorporated | 119.6[a] | 106.5[ab] | 115.4[ab] | 88.7[b] | 7.67 | 0.042 |
| [H] recovery (%) | 84.3 | 75.5 | 84.6 | 77.4 | 4.15 | 0.383 |

[a-b] Within each column, treatment means not sharing the same superscript differ (*P*<0.05).

[1]CTR = 0 mg/ 20g DM of *Ascophyllum nodosum* (ASC) extract; 0.1% = 20 mg/ 20g DM of ASC extract; 0.2% = 40 mg/ 20g DM of ASC extract; 0.3% = 60 mg/ 20g DM of ASC extract.

TGP, Total gas production; $CH_4$, Methane; DOM, digestible organic matter; TVFA, Total volatile fatty acids; [H], Metabolic hydrogen.

## Discussion

This study investigated the effects of an ASC extract on nutrient degradation, ruminal fermentation and $CH_4$ production *in vitro* using the RUSITEC system. Overall, ASC supplementation reduced TGP and $CH_4$ production across both grass- and silage-based diets. In Experiment 1, the inclusion of ASC extract at 0.3% to an 80:20 grass-to-concentrate diet reduced all nutrient degradability parameters, TVFA production, TGP and $CH_4$ production. In Experiment 2, when added to an 80:20 silage-to-concentrate diet, all three inclusion rates of ASC extract (0.3%, 0.4% and 0.5%) reduced TGP and $CH_4$ production, while 0.5% lowered DM and OM degradability. These findings highlight that PT-rich ASC extracts can lower $CH_4$ production *in vitro* through reduced nutrient degradation and fermentation activity, with evidence suggesting that the underlying mechanisms may differ between grass- and silage-based diets.

### Diet degradation

Reductions in $CH_4$ production of 23–34% with the addition of an ASC extract to an 80:20 F:C diet was identified in the current study. The highest inclusion level (0.3% for a grass-based diet and 0.5% for a silage-based diet) of an ASC extract reduced degradation of DM and OM, with reductions in CP and NDF degradation in Experiment 1 (grass-based diet) only. The observed reductions in degradation parameters are likely attributable to the PT content in the seaweed extract, which effect nutrient degradability through antimicrobial activity and nutrient complex formation.

The antimicrobial properties of PT, particularly their effects on fibre-degrading bacteria [21,31], and their nutrient-binding abilities [32] are well-documented. In this study, the higher PT concentrations (408 g/kg DM in Experiment 1; 343 g/kg DM in Experiment 2) likely amplified these effects, reducing nutrient degradation. Similarly, Kunzel [17] reported

**Table 6. Mean effect of treatments on gas production and rumen fermentation parameters in Experiment 2.**

| Parameter | Treatments | | | | SEM | *P*-value |
|---|---|---|---|---|---|---|
| | CTR | 0.3% | 0.4% | 0.5% | | |
| TGP (L/d) | 2.61[a] | 2.04[b] | 1.98[b] | 2.07[b] | 0.124 | 0.003 |
| $CH_4$ (mmol/d) | 8.8[a] | 6.5[b] | 6.4[b] | 6.8[b] | 0.47 | 0.029 |
| $CH_4$ (mmol/g DOM) | 0.69[a] | 0.49[b] | 0.50[b] | 0.54[b] | 0.034 | 0.020 |
| Vessel pH | 7.09 | 7.06 | 7.06 | 7.10 | 0.023 | 0.554 |
| VFA production (mmol/d) | | | | | | |
| TVFA | 46.5 | 46.1 | 45.9 | 51.2 | 2.09 | 0.241 |
| Acetate | 20.9 | 20.7 | 21.4 | 23.8 | 1.03 | 0.159 |
| Propionate | 11.4 | 10.9 | 11.4 | 12.7 | 0.63 | 0.292 |
| Butyrate | 8.10 | 8.38 | 7.82 | 8.87 | 0.410 | 0.329 |
| Iso-butyrate | 0.37 | 0.36 | 0.35 | 0.40 | 0.022 | 0.462 |
| Valerate | 3.75 | 3.86 | 3.37 | 3.74 | 0.159 | 0.188 |
| Iso-valerate | 1.77 | 1.71 | 1.62 | 1.72 | 0.084 | 0.648 |
| Acetate: propionate ratio | 1.85 | 1.88 | 1.88 | 1.88 | 0.040 | 0.909 |
| Estimated H parameters (mmol/d) | | | | | | |
| [H] production | 86.0 | 85.9 | 85.3 | 97.3 | 4.28 | 0.182 |
| [H] incorporated | 77.7 | 67.9 | 67.6 | 78.2 | 3.66 | 0.091 |
| [H] recovery (%) | 90.3[a] | 78.9[b] | 79.5[b] | 78.6[b] | 2.97 | 0.034 |

a-b Within each column, treatment means not sharing the same superscript differ (*P*<0.05).

[1]CTR=0 mg/ 20g DM of *Ascophyllum nodosum* (ASC) extract; 0.1%=20 mg/ 20g DM of ASC extract; 0.2%=40 mg/ 20g DM of ASC extract; 0.3%=60 mg/ 20g DM of ASC extract.

TGP, Total gas production; $CH_4$, Methane; DOM, digestible organic matter; TVFA, Total volatile fatty acids; [H], Metabolic hydrogen.

reductions in all nutrient degradability parameters when dried ASC (5% DM) with a PT concentration of 79 g/kg DM was added to a 60:40 F:C diet. In contrast, Roskam [20] observed no effect on nutrient degradation when ASC was included at 1% DM with a PT concentration of 107 g/kg DM. Likewise, Andreen [33] observed no changes in degradability at 2.5% DM ASC, although PT concentrations were not reported. Belanche [19] included dried ASC at 5% DM but observed reductions in N degradability only, not in fibre degradability. This can be attributed to the lower PT concentration (2.44 g/kg DM) and 50:50 F:C ratio, which affected protein-binding but not carbohydrate-binding [19]. Discrepancies among these studies can be attributed to differences in ASC inclusion level, PT concentrations and F:C ratios. Additionally, PT's show a greater affinity towards fibre-degrading bacteria in comparison to starch-degrading bacteria [21], emphasizing the diet-dependent nature of PT's impacts. These effects vary with microbial population and substrate composition, further highlighting the importance of diet composition determining PT's efficacy.

The grass-based diet in Experiment 1 showed reductions in CP and NDF degradation, while no such effects were observed with the silage-based diet in Experiment 2. This differential response highlights how PT's impacts are modulated by diet composition, with the lower NDF and higher CP content of the grass-based diet enhancing PT's nutrient-binding effects. Additionally, the rapid fermentability of grass may have increased its susceptibility to PT's antimicrobial effects. These findings highlight the diet-dependent nature of PT's impacts, as its effects vary with microbial population and substrate composition.

A reduction in nutrient degradability would result in reduced nutrient and energy availability for ruminants [17]. While a reduction in ruminal CP degradability could be compensated for by an increased post-ruminal degradation, allowing ruminally undegraded protein to be efficiently utilized in the small intestine [34], this is not the case for NDF degradability.

Neutral detergent fibre in the rumen is primarily degraded by fibre-degrading microbes [35], and any reduction in its ruminal degradation is unlikely to be compensated for post-ruminally. Therefore, a reduction in NDF degradability is likely to result in a net loss of digestible energy, potentially impacting animal performance. Optimizing PT inclusion rates is crucial to balance $CH_4$ mitigation with nutrient availability.

## Fermentation parameters

The reduced nutrient degradation in Experiment 1 with the grass-based diet likely decreased substrate availability for fermentation, resulting in lower TGP and VFA production. Tannins can reduce the rate of nutrient digestion and decrease TVFA concentrations [16,36]. In the present study, TVFA and butyrate production were reduced at 0.3% inclusion level compared to CTR, while acetate production was reduced relative to 0.1% treatment only. These effects align with findings by Kunzel [17] and Wang [21], who observed similar reductions in VFA's and gas production due to PT's inhibitory effects on microbial activity and nutrient degradation.

In Experiment 2, while DM and OM degradation decreased, CP and NDF degradation were unaffected. This suggests that fermentation dynamics are influenced by specific microbial populations responsible for fibre and protein degradation. Notably, NDF degradability appears to play a critical role in fermentation as Wang [21] observed that reduced NDF degradability was linked to a reduction in total gas and TVFA production, while Belanche [19] found that reductions in N degradability alone had no effect on the rumen fermentation. These findings align with the results of the current study, where the lack of effect on NDF degradability corresponds with the absence of negative impacts on VFA production.

Metabolic hydrogen produced and incorporated decreased with the 0.3% treatment in Experiment 1, which is in line with the proposed relationship between [H] flux and $CH_4$ production [30]. Despite this, [H] recovery remained consistent, suggesting that the balance between [H] produced and incorporated was maintained, even though microbial activity occurred at a lower level. In Experiment 2, [H] recovered was reduced across all inclusion levels, despite no change in [H] production or VFA production. This indicates that while overall fermentation activity remained constant, a portion of [H] was redirected away from methanogenesis towards alternative sinks or microbial pathways [37]. Such a shift would lower $CH_4$ production irrespective of degradability or TVFA production, consistent with a microbial-driven inhibition mechanism. Previous *in vitro* work with red and brown seaweed has demonstrated this inhibition of methanogen populations [17,38]. Therefore, it is plausible that the ASC extract used in this study impacted methanogenic activity, leading to lower [H] recovery and $CH_4$ production.

Rumen pH plays a vital role in maintaining a stable rumen function because of its effects on microbial fermentation and fermentation end products [39,40]. Rumen pH remained unaffected by treatments in both experiments, likely due to the highly buffered nature of the *in vitro* system, which is designed to prevent significant pH fluctuations. However, differences between experiments were observed, with the grass-based diet in Experiment 1 producing higher VFA concentrations and lower pH than the grass silage-based diet in Experiment 2. These variations are consistent with differences in substrate fermentability, as rapidly fermentable substrate like grass generate more VFA's and lower pH compared to less fermentable silage. Rumen pH is negatively correlated with VFA production [40,41], and hence the greater amount of VFA's seen in Experiment 1 coincide with the lower vessel pH values recorded in that experiment.

## Total gas and methane parameters

There have been promising results involving the use of red seaweed species in reducing $CH_4$ emissions [13,14,42], however, the anti-methanogenic effect of such species can be attributed to the presence of halogenated compounds such as bromoform and dibromochloromethane [43]. The anti-methanogenic potential of brown seaweed species on the other hand, can be aligned towards the concentration of other bioactive compounds such as PT [11]. In the current study, the addition of an ASC extract reduced $CH_4$ production by 23–34% with mechanisms varying between diets. In Experiment 1, reductions in $CH_4$ were linked to decreased nutrient degradation, TGP and VFA production, reflecting a substrate driven

mechanism, where PT-nutrient complex formation reduced fermentable substrate availability and overall fermentation activity, which subsequently lowered TGP and VFA production. This reflects a trade-off between $CH_4$ mitigation and nutrient utilization efficiency. Conversely, in Experiment 2, $CH_4$ reductions occurred without significant effects on degradability or VFA production, although TGP was reduced. This suggests a microbial-driven mechanism, likely involving inhibition of methanogenic populations or redirection of $H_2$ away from methanogenesis towards alternative sinks. Further research is required to better understand these differing mechanisms.

When $CH_4$ production is reduced due to fermentation, it is typically attributed to either reduced overall fermentation activity, reflected by lower VFA and gas production, or shifts in VFA proportions [44,45]. In Experiment 1, as the acetate-to-propionate ratio remained unaffected, the observed reductions in $CH_4$ appear to be primarily due to decreased fermentation activity and lower TGP, rather than altered fermentation pathways. Similar findings were reported by Kunzel [17] and Yergaliyev [18], who observed reductions in fermentation and $CH_4$ production when administering dried ASC at 5% DM and 2.5% DM *in vitro*, respectively. In contrast, studies by Belanche [19] and Roskam [20], found limited effects on fermentation and $CH_4$ production, likely due to lower PT concentrations and ASC dosage rates, respectively.

The observed reduction in [H] recovered in Experiment 2 suggests that not all $H_2$ was accounted for in $CH_4$ or VFA end products. Since methanogens are the primary microbes responsible for converting $H_2$ into $CH_4$ during rumen fermentation [41], it is plausible that PTs inhibited methanogen activity or reduced the efficiency of $H_2$ utilization for methanogenesis. Supporting this, Kunzel [17] reported a reduction in the predominant methanogen genus *Methanobrevibacter* following ASC inclusion *in vitro*, while Belanche [19] observed decreased methanogen and protozoal populations. Moreover, Yergaliyev [18] demonstrated that ASC reduced the relative abundance of archaea, further supporting a microbiome-driven inhibition mechanism. Although microbial populations were not assessed in this study, the consistent reduction in [H] recovery supports a $H_2$-redirection or methanogen-inhibition mechanism underpinning the $CH_4$ reduction observed in Experiment 2.

A limitation of this study is the absence of microbial community analysis, which would have enabled a more comprehensive comparison between experiments and provided greater insight into the mechanisms underpinning the observed effects, particularly in Experiment 2. Future studies should integrate microbial profiling to better elucidate the relationship between ASC supplementation, fermentation dynamics, and methanogen populations.

These findings also highlight the potential of liquid ASC extract as a practical $CH_4$-mitigating feed additive for ruminant production systems. Its water solubility may allow incorporation through on-farm water delivery systems, offering a feasible approach for pasture-based systems. However, further *in vivo* evaluation is required to determine the optimal inclusion rate that achieves sustained $CH_4$ mitigation while maintaining animal performance.

## Conclusion

This study demonstrates that ASC extracts can reduce $CH_4$ production *in vitro* by 23–34%, with the mode of action differing according to the basal diet. In Experiment 1 (grass-based diet), reductions in degradability, TGP and VFA production contributed to lower $CH_4$ output, reflecting a substrate-driven mechanism. In Experiment 2 (silage-based diet), TGP and $CH_4$ production were reduced without effects on degradability or VFA production, suggesting inhibition of methanogenic activity or redirection of $H_2$. Although these findings suggest that ASC extract has potential as a $CH_4$-mitigating additive, comparisons between experiments should be made cautiously due to differences in basal diet and rumen inoculum source, which may have influenced fermentation dynamics and microbial community structure. Future research should incorporate microbial community analysis to better interpret treatment effects across experiments and include long-term *in vivo* trials to evaluate sustained efficacy and impacts on animal performance under practical feeding conditions.

## Supporting information

**S1 Data. Raw data RUSITEC.**
(XLSX)

## Acknowledgments

The authors would like to thank BioAtlantis Ltd for the support provided throughout this study, and to the laboratory staff and postgraduate students at UCD Lyons Farm for the help provided during this study.

## Author contributions

**Conceptualization:** Tommy M. Boland, Zoe C. McKay, John T. O'Sullivan, Kieran J. Guinan, Sivaprakash Balakrishnan, Stafford Vigors.

**Data curation:** Aaron Casey, Alan K. Kelly.

**Formal analysis:** Aaron Casey, Alan K. Kelly.

**Funding acquisition:** Tommy M. Boland, Zoe C. Mckay, Stafford Vigors.

**Investigation:** Aaron Casey.

**Methodology:** Aaron Casey, Alan K. Kelly, Zoe C. McKay, Maria Markiewicz-Keszycka, Marco Garcia-Vaquero, Armin Mirzapour-Koudash, John T. O'Sullivan, Kieran J. Guinan, Sivaprakash Balakrishnan.

**Project administration:** Tommy M. Boland, Stafford Vigors.

**Resources:** Tommy M. Boland, Maria Markiewicz-Keszycka, Marco Garcia-Vaquero, Armin Mirzapour-Koudash, John T. O'Sullivan, Kieran J. Guinan, Sivaprakash Balakrishnan, Stafford Vigors.

**Supervision:** Tommy M. Boland, Stafford Vigors.

**Writing – original draft:** Aaron Casey.

**Writing – review & editing:** Aaron Casey, Tommy M. Boland, Alan K. Kelly, Zoe C. McKay, Stafford Vigors.

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
