## [Decision Letter · Decision Letter 0]

23 Oct 2025

Dear Dr. Vigors,

Thank you for submitting your manuscript to PLOS ONE. After careful consideration, we feel that it has merit but does not fully meet PLOS ONE’s publication criteria as it currently stands. Therefore, we invite you to submit a revised version of the manuscript that addresses the points raised during the review process.

We look forward to receiving your revised manuscript.

Kind regards,

Agung Irawan

Academic Editor

PLOS ONE

Journal Requirements:

3. In the online submission form, you indicated that all relevant data are within the manuscript and its Supporting Information files. Raw data underlying the results are owned in part by BioAtlantis Ltd and cannot be shared publicly due to third-party restrictions. Data may be made available from corresponding author contact details for researchers who meet the criteria for access to confidential data.

5. Please amend your list of authors on the manuscript to ensure that each author is linked to an affiliation. Authors’ affiliations should reflect the institution where the work was done (if authors moved subsequently, you can also list the new affiliation stating “current affiliation:….” as necessary).

Reviewers' comments:

Reviewer's Responses to Questions

**Comments to the Author**

1. Is the manuscript technically sound, and do the data support the conclusions?

Reviewer #1: Partly

Reviewer #2: Yes

2. Has the statistical analysis been performed appropriately and rigorously?

Reviewer #1: Yes

Reviewer #2: Yes

3. Have the authors made all data underlying the findings in their manuscript fully available?

Reviewer #1: Yes

Reviewer #2: Yes

4. Is the manuscript presented in an intelligible fashion and written in standard English?

Reviewer #1: Yes

Reviewer #2: Yes

Reviewer #1: Conclusion should be modified. Please consider that experiment 1 and 2 are not conducted in the same time, the there is difference of rumen donor experiment 1 (slaughter house) and experiment 2 (canulated cattle). Both of variance must be discussed in the Result and Discussion. If the experiment 1 and 2 will be compared, microbiome profiles of both rumen flued should be analysed as co-factor.

Reviewer #2: Dear Authors,

This work has potential value for publication in PLOS ONE, provided that several key issues are addressed. The manuscript would benefit from clearer hypotheses, improved description of experimental design, and enhanced discussion of biological implications.

**Do you want your identity to be public for this peer review?** For information about this choice, including consent withdrawal, please see our Privacy Policy

Reviewer #1: **Yes:** Ahmad Sofyan

Reviewer #2: No

---

## [Author Response · Author response to Decision Letter 1]

19 Jan 2026

Please see responses to the below questions. We appreciate the time taken to review the manuscript

Stafford Vigors

1. We note that your author list was updated during the revision process. In order to add or remove authors or update the order of the author byline after initial submission, we ask that authors complete an Authorship Change Request form. You may review our full authorship change policy and download the Authorship Change Request form here: https://journals.plos.org/plosone/s/authorship#loc-authorship-changes.

Please fill out all required sections of the form. If you are adding or removing more than 2 authors, you can complete multiple forms and submit as many forms as needed to reflect the updates. Please return the form(s) as an attachment by emailing plosone@plos.org or by uploading it as a submission file labeled with the file type ‘Other’. The form will be reviewed by PLOS staff for approval. Should the changes be approved, we will require written confirmation from all authors confirming that they approve of the changes. Please note that if your manuscript is accepted, we will not be able to complete the publication process without the completed form.

Response: I have responded through email regarding the approval for all authors regarding the change. I have uploaded the response email and the author change request form

2. "We note that your Data Availability Statement is currently as follows: "All relevant data are within the manuscript and its Supporting Information files"

If there are ethical or legal restrictions on sharing a de-identified data set, please explain them in detail (e.g., data contain potentially sensitive information, data are owned by a third-party organization, etc.) and who has imposed them (e.g., an ethics committee). Please also provide contact information for a data access committee, ethics committee, or other institutional body to which data requests may be sent. If data are owned by a third party, please indicate how others may request data access."

Response: This file has been uploaded

When you resubmit, please ensure that you provide the correct grant numbers for the awards you received for your study.

In addition, please state what role the funders took in the study. If the funders had no role, please state: "The funders had no role in study design, data collection and analysis, decision to publish, or preparation of the manuscript."

Response: Funding information has now been included

Kindly include this amended Funding disclosure statement in your cover letter; we will change the online submission form on your behalf.

Response: Funding information has now been included on the online system

---

## [Editor Report · Decision Letter 1]

11 Feb 2026

Dear Dr. Vigors,

Thank you for submitting your manuscript to PLOS ONE. After careful consideration, we feel that it has merit but does not fully meet PLOS ONE’s publication criteria as it currently stands. Therefore, we invite you to submit a revised version of the manuscript that addresses the points raised during the review process.

We look forward to receiving your revised manuscript.

Kind regards,

Agung Irawan

Academic Editor

PLOS One

Journal Requirements:

Additional Editor Comments:

I see that authors have made revisions to their manuscript but I do not see the authors' responses to each comment from the reviewer. Please make sure that you submit your revision along with point-by-point response to the reviewers' comments. Thank you.

---

## [Author Response · Author response to Decision Letter 2]

11 Feb 2026

There have been two rounds of reviewer/editor comments. I have attached response to both in the submission and also included both below.

Cover letter revisions for manuscript 1

Manuscript number: PONE-D-25-49338

Title: Evaluating the effect of an Ascophyllum nodosum extract on diet degradability, ruminal fermentation and methane production using the rumen simulation technique (RUSITEC)

Dear Agung Irawan,

Thank you for the response to the above submission. Please see our comments below in

detail.

Yours sincerely,

Dr. Stafford Vigors.

Author’s comment: The authors thank both reviewers for their time in carefully reviewing the manuscript. Please see below our responses and the edits/changes to the manuscript highlighted in yellow.

Reviewer #1: PONE-D-25-49338

Please revise the title: Evaluation of Ascophyllum nodosum supplementation on feed degradability, ruminal fermentation and methane production using Rusitec

Corrected and highlighted lines 1-3

Line 52, what is the reason and significance, author use A nodosum? What is the mutual advantage compared to the other seaweed?

Thank you for the comment. Ascophyllum nodosum was selected as it is a native and abundantly available brown seaweed in Ireland that does not contain bromoform yet still exhibits anti-methanogenic potential due to its high phlorotannin content. Rewritten to further clarify in lines 49-54.

Line 114, Where are rumen come from? Please describe breed, sex, BW, age.

The rumen fluid for experiment 1 was collected from the rumens of three slaughter Angus x Friesian steers and has been added and highlight on line 121-122.

Line 116, Please describe sex, BW, age of cattle.

Sex, BW and age of the cattle has been added and highlighted on line 124-125.

Line 120, Ascophyllum nodosum highlighted.

Changed to ASC and highlighted in line 129.

Line 170, explain DC abbreviation.

“DC” is part of the model name designated by the manufacturer (Shinagawa Corp.; Tokyo, Japan) and does not represent an abbreviation.

Line 173, explain TGP abbreviation.

TGP is abbreviated for the first time and highlighted on line 62.

TGP is abbreviated again and highlighted for the formula on line 185.

Line 184, what is the method for N determination?

AOAC 2005c; method 990.03 highlighted in line 195.

Line 204, please “(gallic acid (TPC)” re-write clearly.

Rewritten and highlighted in line 214-215.

Line 220-221, Please rewrite “1,800 g” add “x” between 1,800 and g.

Rewritten and highlighted on line 230-231 and 233 including “×”

Line 226, add information method/instrument type e.g. gas chromatography (456-GS Scion instrument, Scotland, 227 UK)?

Method/instrument clarified and highlighted on line 236-238.

Line 237, What is PDIFF?

PDIFF is the option of the LSMEANS statement in proc mixed. This has been further clarified and highlighted in line 247-248.

Conclusion

Conclusion should be modified. Please consider that experiment 1 and 2 are not conducted in the same time, there is difference of rumen donor experiment 1 (slaughtered house) and experiment 2 (cannulated cattle). Both variances must be discussed in the results and discussion.

If the experiment 1 and 2 will be compared, microbiome profiles of both rumen fluid should be analysed as co-factors.

We appreciate the reviewer’s valuable feedback. The conclusion has been modified to acknowledge that differences in basal diet and rumen inoculum source may have influenced fermentation dynamics and microbial community structure, and that direct comparison between experiments should therefore be made with caution. The need for future microbial community analysis has also been emphasized to better interpret results between experiments. This has been added and highlighted in line 433-444.

Reviewer #2: PONE-D-25-49338

General Comments

This work has potential value for publication in PLOS ONE, provided that several key issues are addressed. The manuscript would benefit from clearer hypotheses, improved description of experimental design, and enhanced discussion of biological implications. Specific comments by section are outlined below.

Specific Comments

Abstract

Line 27-28: Summarize results concisely. Quantify methane reduction (e.g., “Methane decreased by 12–18% relative to control”). Conclude with practical significance (e.g., “These results indicate that A. nodosum extract may serve as a natural methane mitigation strategy under controlled fermentation conditions.”)

Results in the abstract have been summarized more concisely, with quantified methane reduction percentages now included and highlighted on lines 18–20 and 23–25. The concluding sentence has been revised to emphasize the practical significance of the findings, as shown on lines 26–29.

Introduction

Line 54-56: The text states that phlorotannins affect the activity of fiber-degrading microorganisms. However, in ruminants, these fiber-degrading microbes play a crucial role in feed digestion and overall nutrient utilization. Include discussion of phlorotannins and their possible role in modulating rumen fermentation.

We appreciate the reviewer’s comment. The section has been revised to provide further discussion on the role of phlorotannins in modulating rumen fermentation and their potential impact on nutrient utilization on lines 56-61.

Line 73-76: Add a clearly stated hypothesis. For instance, “A. nodosum extract would reduce

methane emissions without compromising fermentation efficiency.”

Hypothesis sentence has been more clearly stated and highlighted in lines 77-80.

Provide rationale for selecting grass- vs silage-based diets to highlight differences in fermentability and baseline methane output.

We appreciate the reviewer’s comment. Additional clarification has been provided in the materials and methods section on lines 108-110 to explain that the use of grass and silage diets aimed to minimise microbial adaptation time while also enabling evaluation of Ascophyllum nodosum supplementation across forages differing in fermentability and baseline CH4 potential.

Results

Present summarized graphs (e.g., bar charts for methane and total gas production) for improved readability.

We appreciate the reviewer’s suggestion. However, we feel that the current presentation of results in the tables provides a clear and comprehensive summary of the data, including measures of variation and statistical significance. Adding graphs for these parameters would duplicate the information already presented in the tables without improving clarity. Therefore, we have opted to retain the existing format for consistency and conciseness.

Report % change or relative differences for methane reduction.

Methane reduction % have been added to the results section in line 284-287.

Discuss whether methane reduction corresponds to lower total gas or degradability.

We thank the reviewer for this helpful comment. This has now been clarified in the revised Discussion section in lines 392-400. In Experiment 1, the reduction in CH4 production was associated with decreased nutrient degradability, total gas, and VFA production, indicating a substrate-driven mechanism whereby lower fermentable substrate availability limited overall fermentation activity. In Experiment 2, CH4 reductions occurred without changes in degradability or VFA production, suggesting a microbial-driven mechanism, potentially involving inhibition of methanogenic activity or redirection of hydrogen away from methanogenesis.

Present molar proportions of VFAs (acetate, propionate, butyrate) to support interpretation of

fermentation patterns.

We appreciate the reviewer’s suggestion. However, in the present study, no differences were observed in the molar proportions of individual VFAs among treatments. Therefore, we feel that inclusion of these data would not enhance interpretation of the results. As in other published in vitro studies (Belanche et al., 2016; Roskam et al., 2022; Kunzel et al., 2022), VFA production has been expressed as mmol/d to represent the total fermentation products collected over the 24 h incubation period. This approach provides a direct measure of total substrate fermentation within the 24 h incubation timeframe, which we consider most reflective of the overall fermentation profile under in vitro conditions. However, if requested we can add the molar proportion data

Ensure that all tables include p-values or superscripts for statistical differences.

P-values and superscripts are included in all tables where statistical analyses were performed. The first two tables present only the chemical composition of the feed ingredients and seaweed extract, therefore, statistical comparisons were not applicable.

Discussion

Line 287-292: Begin with a brief summary of main findings before interpretation.

A brief summary of the main findings before interpretation has been added and highlighted in lines 292-302.

Line 401: Discuss potential mechanisms — e.g., phlorotannin effects on methanogens, hydrogen flow, or protozoal populations.

The potential mechanistic effects of PTs have been highlighted in this section. The substrate-driven mechanisms are highlighted in lines 392-396. The microbial-driven mechanisms are highlighted in lines 397-400.

Acknowledge the limitation of lacking microbial community data.

A paragraph on the limitation of lacking microbial community data is highlighted in lines 422-426.

Include updated literature (2020–2024) on seaweed extracts and methane mitigation.

Several updated references from 2020-2025 have been included throughout the manuscript including Sofyan (2022), Yergaliyev (2025), Martins (2024), Almeida (2021), Beauchemin (2022), Stefenoni (2021), Ahmed (2022), Andreen (2025), Kelln (2021), Alvarez-Hess (2024) and Ungerfeld (2023).

Discuss practical application and potential use in in vivo feeding systems.

A paragraph on the practical application and use of water-based systems in vivo is highlighted in lines 427-431.

Conclusion

Conclude with future research directions (e.g., microbiome analysis, long-term feeding trials).

We thank the reviewer for this suggestion. The conclusion has been revised to include future research directions, highlighting the need for microbial community analysis and long-term in vivo trials to confirm the mechanisms observed and assess sustained efficacy under practical feeding conditions. Highlighted in lines 433-444.

Table

The ash content values in Table 1 (Chemical composition) should be rechecked for accuracy.

Ash content recorrected in Table 1.

Please see responses to the below questions. We appreciate the time taken to review the manuscript

Stafford Vigors

1. We note that your author list was updated during the revision process. In order to add or remove authors or update the order of the author byline after initial submission, we ask that authors complete an Authorship Change Request form. You may review our full authorship change policy and download the Authorship Change Request form here: https://journals.plos.org/plosone/s/authorship#loc-authorship-changes.

Please fill out all required sections of the form. If you are adding or removing more than 2 authors, you can complete multiple forms and submit as many forms as needed to reflect the updates. Please return the form(s) as an attachment by emailing plosone@plos.org or by uploading it as a submission file labeled with the file type ‘Other’. The form will be reviewed by PLOS staff for approval. Should the changes be approved, we will require written confirmation from all authors confirming that they approve of the changes. Please note that if your manuscript is accepted, we will not be able to complete the publication process without the completed form.

Response: I have responded through email regarding the approval for all authors regarding the change. I have uploaded the response email and the author change request form

2. "We note that your Data Availability Statement is currently as follows: "All relevant data are within the manuscript and its Supporting Information files"

If there are ethical or legal restrictions on sharing a de-identified data set, please explain them in detail (e.g., data contain potentially sensitive information, data are owned by a third-party organization, etc.) and who has imposed them (e.g., an ethics committee). Please also provide contact information for a data access committee, ethics committee, or other institutional body to which data requests may be sent. If data are owned by a third party, please indicate how others may request data access."

Response: This file has been uploaded

When you resubmit, please ensure that you provide the correct grant numbers for the awards you received for your study.

In addition, please state what role the funders took in the study. If the funders had no role, please state: "The funders had no role in study design, data collection and analysis, decision to publish, or preparation of the manuscript."

Response: Funding information has now been included

Kindly include this amended Funding disclosure statement in your cover letter; we will change the online submission form on your behalf.

Response: Funding information has now been included on the online system

---

## [Editor Report · Decision Letter 2]

15 Feb 2026

Evaluation of Ascophyllum nodosum extract supplementation on feed degradability, ruminal fermentation and methane production using the rumen simulation technique (RUSITEC)

PONE-D-25-49338R2

Dear Dr. Vigors,

We’re pleased to inform you that your manuscript has been judged scientifically suitable for publication and will be formally accepted for publication once it meets all outstanding technical requirements.

Kind regards,

Agung Irawan

Academic Editor

PLOS One
---

## [Editor Report · Acceptance letter]

PONE-D-25-49338R2

PLOS One

Dear Dr. Vigors,

I'm pleased to inform you that your manuscript has been deemed suitable for publication in PLOS One. Congratulations! Your manuscript is now being handed over to our production team.

Kind regards,

on behalf of

Dr. Agung Irawan

Academic Editor

PLOS One